# Cloning and Characterization of Two Novel *PR4* Genes from *Picea asperata*

**DOI:** 10.3390/ijms232314906

**Published:** 2022-11-28

**Authors:** Weidong Zhao, Lijuan Liu, Chengsong Li, Chunlin Yang, Shujiang Li, Shan Han, Tiantian Lin, Yinggao Liu

**Affiliations:** National Forestry and Grassland Administration, Key Laboratory of Forest Resources Conservation and Ecological Safety on the Upper Reaches of the Yangtze River, Forestry Ecological Engineering in the Upper Reaches of the Yangtze River, Key Laboratory of Sichuan Province, College of Forestry, Sichuan Agricultural University, Chengdu 611130, China

**Keywords:** *Picea asperata*, PR4 protein, prokaryotic expression, inclusion body renaturation, antifungal activity, subcellular localization, overexpression

## Abstract

Pathogenesis-related (PR) proteins are important in plant pathogenic resistance and comprise 17 families, including the PR4 family, with antifungal and anti-pathogenic functions. PR4 proteins contain a C-terminal Barwin domain and are divided into Classes I and II based on the presence of an N-terminal chitin-binding domain (CBD). This study is the first to isolate two *PR4* genes, *PaPR4-a* and *PaPR4-b*, from *Picea asperata*, encoding PaPR4-a and PaPR4-b, respectively. Sequence analyses suggested that they were Class II proteins, owing to the presence of an N-terminal signal peptide and a C-terminal Barwin domain, but no CBD. Tertiary structure analyses using the Barwin-like protein of papaya as a template revealed structural similarity, and therefore, functional similarity between the proteins. Predictive results revealed an N-terminal transmembrane domain, and subcellular localization studies confirmed its location on cell membrane and nuclei. Real-time quantitative PCR (RT-qPCR) demonstrated that *PaPR4-a* and *PaPR4-b* expression levels were upregulated following infection with *Lophodermium piceae.* Additionally, *PaPR4-a* and *PaPR4-b* were induced in *Escherichia coli*, where the recombinant proteins existed in inclusion bodies. The renatured purified proteins showed antifungal activity. Furthermore, transgenic tobacco overexpressing *PaPR4-a* and *PaPR4-b* exhibited improved resistance to fungal infection. The study can provide a basis for further molecular mechanistic insights into PR4-induced defense responses.

## 1. Introduction

The eukaryotic immune system has the unique ability to recognize invading pathogens and rapidly initiate an appropriate defense response [1]. Plants are challenged with various pathogenic infections during growth, which hinders their growth and adaptability [2]. Plants have developed two immune responses during their evolution: the non-specific immune response, which is induced by pathogenic associated molecular patterns (PTIs) [3], and the specific immune response, which is more potent and is induced by effector proteins (ETIs) [4]. Pathogenesis-related (PR) proteins have been widely recognized for their importance in plant-pathogen interactions. Numerous PR proteins have been identified and reported, which can be divided into 17 families based on their primary structure, serological relationships, and biological activities [5].

As a member of the PR protein families, the PR4 protein family is characterized by the presence of a C-terminal Barwin domain containing six cysteine residues that form three disulfide bonds (Cys31-Cys63, Cys52-Cys86, and Cys66-Cys123). The Barwin domain also has the ability to bind sugar moieties [6,7]. Most PR4 proteins have some extended residues at the C-terminal, which are necessary for the vacuolar targeting of PR4 proteins [8]. The PR4 family can be divided into Classes I and II, based on the presence or absence of an N-terminal chitin-binding domain (CBD) containing cysteines. Class I proteins contain a CBD and include the CBP20 protein of *Nicotiana tabacum* [9] and the HEl protein of *Arabidopsis* sp. [10], among others. Class II proteins, including the PgPR4 protein of *Panax ginseng* [11] and the wheatwin protein of wheat [12], lack a CBD. Although numerous studies have reported the identification of PR proteins, their specific roles in plant defense responses remain to be thoroughly investigated. Existing studies have demonstrated that PR4 proteins have RNase and DNase activities and can inhibit spore germination and the growth of fungal mycelia [13,14,15]; however, the specific molecular mechanism underlying the PR4-induced defense response remains to be elucidated. To date, proteins of the PR4 family have been identified in a variety of dicotyledons. For instance, a study confirmed the positive role of the PR4 protein encoded by *VvPR4b* gene of *Vitis vinifera* in disease resistance and reported that compared with wild type plants, transgenic plants are susceptible to downy mildew following *VvPR4b* knockout [16]. Additionally, the expression of *LhSorPR4a* and *LhSorPR4b* in the monocotyledonous plant, lily, is induced by methyl jasmonate (MeJA) and ethethrel (ETH), indicating the important role of PR4 proteins in the disease resistance of lily plants [17]. Another study reported that the expression levels of genes encoding PR4 proteins are upregulated in both seedlings and adult gramineous wheat plants following infection with the stripe rust fungus [18]. Studies on gymnosperms have reported genes of the PR4 family, namely, *PmPR4a1*, *PmPR4a2*, and *PmPR4b1*, in Douglas fir, and their expression levels have been shown to be upregulated following infection with *Phellinus sulphurascens* [19].

As an evergreen arbor, *Picea asperata*, belonging to gymnosperum, is an important plant for afforestation in southwest China which plays significant roles in the water and soil conservation of the area and maintenance of regional ecological balance. Needle cast disease was recently detected in several *Pi. asperata* plantations and is spreading to other regions. It was reported that needle cast disease can cripple the growth of *Pi. asperata*, resulting in sparse branches and leaves, and severe infections can even cause plant death. *Lophodermium piceae* is an endophytic fungus that colonizes the needles of *Pi. asperata* and has a very rich variety. UPTOHERE The ascospores of *Lo. piceae* can infect the needles of spruce [20]. We believe that *Lo. piceae* is one of the major pathogens responsible for the needle cast disease of *Pi. asperata*. Previous studies have described the isolation of *Lo. piceae* from the Sichuan Province [21]. Owing to the important roles of the PR4 protein family in plant disease resistance, it is necessary to study their roles in the defense system of *Pi. asperata*. To this end, two genes of the PR4 family were identified and cloned from *Pi. asperata* in this study, including *PaPR4-a* (GenBank ID: OL617012) and *PaPR4-b* (GenBank ID: OL617013). The genes were subsequently expressed in a prokaryotic system and the antifungal activity of the encoded proteins was detected. Their site of action was determined by analyzing the subcellular localization; additionally, the expression levels of *PaPR4* in different stages of disease progression were analyzed. To date, this study is the first to report the identification of PR4 proteins of *Pi. asperata*.

There are several *PR* genes in host plants which are activated following infection with fungi. These genes have been isolated, characterized and cloned to express transgenic resistance to fungal pathogens [22]. These antifungal proteins have a distinct role in the natural plant defense system because they are activated during hypersensitive response (HR) and systemically acquired resistance (SAR) [23] and their expression is induced by topical application of phytohormones or exposure to biotic or abiotic stress conditions [24,25]. When plants are infected with pathogens, the defense system of plants will produce a large number of reactive oxygen species (ROS). The production and accumulation of ROS can lead to membrane lipid peroxidation and loss of membrane differential permeability, thus causing a series of physiological and biochemical changes. Major ROS scavenging enzymes in plant include superoxide dismutase (SOD; EC 1.15.1.1), catalase (CAT; EC 1.11.1.6) and peroxidase (POD; EC 1.11.1.7). SOD, a key enzyme protecting cells from oxidative stress, can dismutate superoxide radicals to H_2_O_2_, and CAT and POD can decompose H_2_O_2_ into water and oxygen [26,27,28,29,30]. The activities of the three ROS scavenging enzymes can indirectly reflect the metabolic changes of reactive oxygen species in plants, which are closely related to the disease resistance of plants. When plants such as *Lilium*, kumquat (*Fortunella margarita* (Lour.) Swingle) and *Vitis vinifera* L. were infected by pathogens, their ROS scavenging enzymes activities were upregulated, and the ROS enzyme activities of resistant plants were stronger [31,32,33,34,35,36,37]. The function of the *PaPR4-a* and *PaPR4-b* was further investigated by overexpression experiment. However, regeneration system of tissue culture through callus has not been established due to the growth characteristics of the *Pi. asperat* itself, the *PaPR4-a* and *PaPR4-b* were overexpressed in *Nicotiana benthamiana* respectively and their biological function in defending against *Lo. Piceae* were analyzed. The disease resistance of transgenic plants was determined by measuring the activities of three ROS scavenging enzymes and the relative expression levels of *PaPR4-a* and *PaPR4-b*. This study provides a basis for further exploration of the role of PR4 proteins in plant disease resistance and a new strategy for disease resistance breeding.

## 2. Results

### 2.1. Cloning and Bioinformatics Analyses of PaPR4-a and PaPR4-b Genes

Two genes, *PaPR4-a* and *PaPR4-b* were cloned using the cDNA of *Pi. asperata*; the primers used for cloning the full-length genes are enlisted in Appendix A. The complete sequences of the *PaPR4-a* and *PaPR4-b* genes contained 471 and 447 bp, respectively, and encoded proteins of 157 and 149 amino acids, respectively (Appendix A). The results of bioinformatics analyses demonstrated that both PaPR4-a and PaPR4-b had conserved Barwin domains and belonged to the PR4 protein family. Additionally, it was found that they had N-terminal signal peptides of 36 and 29 amino acids, respectively. The theoretical molecular weights of PaPR4-a and PaPR4-b were 38.383 and 37.094 kDa, respectively, and the theoretical isoelectric point (pI) of both the proteins was 5.21. Analysis of the amino acid composition revealed that both proteins contained a higher proportion of glycine, alanine and threonine residues. The predicted stability index of PaPR4-a was 39.34, indicating structural stability. However, the predicted stability index of PaPR4-b was 56.54, indicating that the protein was structurally unstable. Prediction of protein hydrophilicity revealed that both PaPR4-a and PaPR4-b were hydrophobic in nature.

The sequence homology of PaPR4-a and PaPR4-b proteins with other PR4 proteins was determined with the BLAST algorithm of National Center for Biotechnology Information (NCBI). The results demonstrated that PaPR4-a had highest sequence similarity (99.20%) with a protein of the Barwin family (ABK23104.1) from *Picea sitchensis* and 96% sequence similarity with a PR4 protein of *Pseudotsuga menziesii* (AFD50744.1). PaPR4-b had highest sequence similarity (93.92%) with a PR4 protein from *Ps*. *menziesii* (AFD50743.1). The protein sequences of PaPR4-a and PaPR4-b were aligned to sequences of Class I proteins; the results demonstrated the absence of a CBD, indicating that PaPR4-a and PaPR4-b were Class II PR4 proteins. The protein sequences of PaPR4-a and PaPR4-b also lacked signal peptides for vacuolar targeting (Figure 1).

Analysis of the secondary and tertiary structures of PaPR4-a and PaPR4-b revealed that the number and location of α-helices, η-helices, β-chains and strict β-turns (TTT) were similar between the two proteins, and both PaPR4-a and PaPR4-b contained six cysteine residues that formed three disulfide bonds (Figure 2). The structural similarity between the proteins led us to hypothesize that both PaPR4-a and PaPR4-b have similar functions.

In order to analyze the evolutionary relationship of PaPR4-a and PaPR4-b with the PR4 proteins of other plants, a phylogenetic tree was constructed using the adjacency method, including the PR4 proteins of angiosperms (monocotyledons and dicotyledons), bryophytes, ferns and gymnosperms (Figure 3). The results revealed that PaPR4-a and PaPR4-b were phylogenetically closely related to PR4 proteins from gymnosperms. We also observed that the phylogenetic relationships between the PR4 proteins were consistent with those of plants, indicating that the PR4 protein sequences were highly conserved during evolution.

### 2.2. Prokaryotic Expression of PaPR4-a and PaPR4-b in E. coli

The pET-32a-PaPR4-a and pET-32a-PaPR4-b recombinant plasmids were successfully transformed into *E. coli* BL21 (DE3) cells. T7-F and T7-R (Appendix A) are used as validation primers to validate positive plasmids. The plasmid maps and results of identification are provided in Appendix A. In order to enhance the expression of PaPR4-a and PaPR4-b proteins, the optimal concentration of isopropyl-β-D-thiogalactoside (IPTG), duration of induction and induction temperature were determined. The results demonstrated that the concentration of IPTG and duration of induction did not affect the expression of PaPR4-a and PaPR4-b. The expression of PaPR4-a and PaPR4-b proteins was highest at 30 °C. The experiments on solubility detection revealed that PaPR4-a and PaPR4-b existed in inclusion bodies (Figure 4 and Figure 5). As inclusion bodies are inactive [38], the PaPR4-a and PaPR4-b proteins were isolated from the inclusion bodies and purified prior to conducting the antifungal activity tests.

By renaturation and purification of the inclusion bodies, we found that the recombinant protein PaPR4-a and PaPR4-b were soluble in an eluent containing 6 M and 8 M urea. (Figure 4 and Figure 5). The purified recombinant protein can be used for subsequent antifungal activity experiments. We attempted to completely remove the urea and renature the recombinant proteins; however, the process proved to be unsuccessful. Therefore, 6 M urea was present in all the eluents of the recombinant proteins.

### 2.3. Antifungal Activity of Recombinant PaPR4-a and PaPR4-b Proteins

In order to detect the antifungal activity of PaPR4-a and PaPR4-b, 200 μL of purified protein was extracted. A *Lo*. *piceae* fungus cake with a diameter of 0.8 cm was placed in the purified protein solution for 24 h at 25 °C. Mycelial morphology was observed under a microscope at a magnification of 40×. We observed that the mycelial morphology of the experimental group was altered, and the fungal inclusion bodies appeared to be condensed compared to those of the control group (Figure 6). The mycelium of fungi plays an important role in the growth and infection of fungi by absorbing nutrients. The abnormal morphology of mycelia indicated directly reflects the positive role of PaPR4-a and PaPR4-b in the process of plant disease resistance.

### 2.4. Subcellular Localization of PaPR4-a and PaPR4-b

The EGFP-PaPR4-a and EGFP-PaPR4-b recombinant plasmids were transformed into *Agrobacterium*-competent GV3101 cells. BamHI-yz-F and BamHI-yz-R validation primers (Appendix A) were used to amplify target bands to detect positive plasmids. The plasmid maps and results of identification are provided in Appendix A. The green fluorescence signal of the empty pCAMBIA1300-EGFP-MCS was visible in the cells of tobacco leaves under confocal microscopy. The red fluorescence was the membrane Marker and nucleus Marker. The results demonstrated that the fluorescence signals due to the membrane Marker and nucleus Marker corresponded to the recombinant PaPR4-a and PaPR4-b proteins (Figure 7), indicating that PaPR4-a and PaPR4-b were located on the nuclei and cell membrane.

### 2.5. Quantitative Analysis of Gene Expression

The expression levels of *PaPR4-a* and *PaPR4-b* were determined at different times following the infection of *Pi. asperata* with *Lo*. *piceae* using Real-time quantitative PCR (RT-qPCR); The elongation factor-1 alpha (EF) (GenBank ID: AJ132534.1) and the translation initiation factor 5A (TIF) (GenBank ID: DR448953.1) were used as the internal reference genes, and samples at 0 h of infection were used as the control. The results demonstrated that the expression levels of *PaPR4-a* and *PaPR4-b* were significantly upregulated in the early stage of infection and finally stabilized at a high level (Figure 8), indicating that PaPR4-a and PaPR4-b were involved and played an active role in the defense response of *Pi. asperata*.

### 2.6. Functional Verification of PaPR4-a and PaPR4-b

The expression level of *PaPR4-a* and *PaPR4-b* in twelve positive transgenic plants were determined using Real-time quantitative PCR (RT-qPCR) with the primers PaPR4-a-Fq/PaPR4-a-Rq and PaPR4-b-Fq/PaPR4-b-Rq (Appendix A). The plasmid maps and results of identification are provided in Appendix A. Finally, six independent transgenic plants (OEa2, OEa4, OEa5 and OEb1, OEb2, OEb4) were chosen for further study (Figure 9a). Having demonstrated that transgenic plants harbored higher expression level of *PaPR4-a* and *PaPR4-b* than wild-type (WT) plants, we assessed the disease phenotypes for leaves inoculated with *Lo. piceae* from transgenic lines. Seven days after the infection, the leaves exhibited less symptoms of curls and yellowish disease spots than WT leaves (Figure 9b). These results indicated that the transgenic plants were more resistant to disease than the wild-type plants.

The role of *PaPR4-a* and *PaPR4-b* in disease resistance was further verified by detecting reactive oxygen species (ROS) scavenging enzymes activities and the relative expression of *PaPR4-a* and *PaPR4-b* after infection. The results showed that the activities of ROS scavenging enzymes in both transgenic and WT plants increased after *Lo. piceae* inoculation. In contrast, the activities of superoxide dismutase (SOD), peroxidase (POD) and catalase (CAT) in transgenic plants were about 2.0-fold, 2.5-fold and 1.8-fold higher than those in WT plants, respectively (Figure 9c). The increased activities of ROS scavenging enzymes can remove more reactive oxygen radicals and H_2_O_2_, which can protect the integrity of the cell membrane. In addition, the relative expression levels of *PaPR4-a* and *PaPR4-b* in transgenic plants were significantly increased by 6–7 times after infection (Figure 9d), indicating that the enhanced resistance of transgenic plants to *Lo. piceae* may be associated with upregulated expression of *PaPR4-a* and *PaPR4-b*, further demonstrating the positive role of *PaPR4-a* and *PaPR4-b* in plant disease resistance.

## 3. Discussion

In this study, two novel genes of the PR4 protein family, namely, *PaPR4-a* and *PaPR4-b*, were identified and cloned from *Pi. asperata*. To the best of our knowledge, this study is the first to report the identification of PR4 proteins from *Pi. asperata*. Comparison of the amino acid sequences of the proteins encoded by *PaPR4-a* and *PaPR4-b* with those of other PR4 proteins revealed that both proteins had a conserved Barwin domain but lacked a chitin-binding domain (CBD), indicating that they were Class II PR4 proteins. Amino acid sequence analyses revealed that both the proteins contained an N-terminal signal peptide and a transmembrane region. The results of subcellular localization studies also revealed that PaPR4-a and PaPR4-b were located on the cell membrane and nuclei. Analysis of the antifungal activity revealed that PaPR4-a and PaPR4-b affected the mycelial morphology and inclusion bodies of *Lo*. *piceae*. These results suggested that *PaPR4-a* and *PaPR4-b* play a crucial role in the defense response of *Pi. asperata.*

The functions and enzymatic activities of proteins are determined by transferring and expressing the target gene into *E. coli* or other prokaryotic cells for obtaining the heterologous target protein; however, *E. coli* is one of the alternative systems [39,40], because this approach has the additional advantages of high efficiency, simple operation protocol and high stability [41,42,43,44]. In order to detect the properties of the recombinant PaPR4-a and PaPR4-b proteins, we constructed recombinant expression vectors and transformed them in *E. coli* BL21 (DE3) cells. The growth of *E. coli* cells would be affected by the N-terminal signal peptides of PaPR4-a and PaPR4-b, which would in turn result in the failure of expression of the recombinant proteins [45,46]. We therefore linked the nucleotide sequence devoid of the region encoding the signal peptide to the pET-32a vector and successfully expressed the recombinant vector. SDS-PAGE revealed that the molecular weight of the encoded protein was approximately 29 KDa (Figure 4 and Figure 5). The conditions of induction were optimized for enhancing the expression of the recombinant PaPR4-a and PaPR4-b proteins. As expected, the recombinant PaPR4-a and PaPR4-b proteins existed in the form of inclusion bodies (Figure 4 and Figure 5). Although the solubility of recombinant proteins can be improved by lowering the temperature [47,48], we observed that the recombinant PaPR4-a and PaPR4-b proteins continued to exist in the form of inclusion bodies even when the induction temperature had been lowered. The dissolution of inclusion bodies and renaturation of the purified recombinant proteins were subsequently performed. The recombinant PaPR4-a and PaPR4-b proteins were finally obtained from the eluents containing 8 M, 6 M and 4 M urea (Figure 4 and Figure 5). Analysis of the antifungal potential of the recombinant proteins revealed that the recombinant PaPR4-a and PaPR4-b proteins affected the fungal mycelial morphology and inclusion bodies compared to those of the control group (Figure 6).

Bioinformatics analyses revealed that PaPR4-a contained a typical transmembrane domain (residues 13–35) and a signal peptide (residues 1–36). PaPR4-b was predicted to contain an N-terminal transmembrane domain (residues 5–27) and a signal peptide (residues 1–29). Previous studies have demonstrated that the hydrophobic amino acid sequences at the N-termini of proteins commonly possess signal peptides [49,50]. We therefore speculated that PaPR4-a and PaPR4-b are secretory proteins that function across membranes. The results of subcellular localization studies in tobacco leaves also demonstrated that they were localized on the cell membrane and nuclei (Figure 7). The experimental results obtained herein can serve as a reference for future studies on the PR4 protein family.

The expression levels of multiple PR genes have been shown to increase and accumulate in plants following infection with pathogenic bacteria [51,52,53], which may be related to the PR protein-induced disease resistance. Quantitative analysis of the expression of *PaPR4-a* and *PaPR4-b* revealed that their expression levels were upregulated following infection, and that the encoded PaPR4-a and PaPR4-b proteins had accumulated (Figure 8), thus confirming their role in disease resistance. It has been demonstrated that proteins of the PR4 family usually possess chitinase activity along with DNase or RNase activity. For instance, the FaPR4 protein of *F*. *awkeotsang* has weak chitinase activity along with RNase activity, and the FaPR4 protein contains a CBD unlike the FaPR4-C protein, which lacks a CBD. The antifungal activity of FaPR4 is more potent than that of FaPR4-C, and FaPR4 retains the RNase activity following heat treatment, while the FaPR4-C protein loses these activities following heat treatment, suggesting that the presence of a CBD improves the thermal stability of FaPR4 [54]. It has been reported that the AtHEL protein of *Arabidopsis thaliana* is a Class I PR protein and possesses a CBD but lacks chitinase activity [10]. Therefore, there is no direct relationship between the chitinase activity and presence of CBD in PR4 proteins. Numerous studies have investigated the DNase and RNase activities of PR4 proteins. The AtHEL and FaPR4 proteins are Class I proteins and possess RNase and antifungal activities [10]. The Class II PR4 protein, LrPR4, from *Lycoris radiata* has RNase activity, which can degrade the RNA of *Ly. Radiata* [14]. The CcPR4 protein of *C*. *chinense* and the CcPR4 protein of *Theobroma cacao* are Class II PR4 proteins with RNase and DNase activities [55,56]. Previous studies have demonstrated that ribonucleases are secreted into the intercellular fluid following induction and may play a protective role by degrading the DNA of pathogens [57]. Investigation of the antifungal mechanism of PaPR4-a and PaPR4-b revealed the absence of DNase or RNase activities; however, some studies have analyzed the tertiary structure of the wheatwin1 protein and proposed that wheatwin1 can inhibit fungal growth and play a role in disease resistance by destroying the cell membrane of pathogens [58]. The OsPR-4b protein of rice has no DNase and RNase activity but has antifungal activities [59]. We therefore believe that the antifungal activity of PR4 proteins could be mediated via another unique mechanism.

The antifungal activity of recombinant proteins is just one piece of evidence that *PR* genes are involved in plant disease resistance response. The function of *PR* genes should be further studied by transgenic method. The efficiency of *PR* genes in transgenic approaches to obtain pathogen resistance is well documented [60,61]. Furthermore, significant constitutive expression of *PRs* in transgenic plants overexpressing *PR* genes accompanied by increased resistance to pathogens. The *PR* genes have been reported to have a defensive effect against specific pathogens or oobacteria in genetically modified plants such as tobacco, tomatoes and potatoes [62,63,64,65,66]. The reactive oxygen species (ROS) burst in response to biotic stress has a protective effect when plants are infected with pathogens, as evidenced by studies showing that some ROS function as secondary messengers of signal transduction pathways controlling pathogen defense responses [67,68]. However, excessive ROS causes serious damage, and plants tightly regulate ROS production and detoxification [69]. Based on these reports, we hypothesize that *PaPR4-a* and *PaPR4-b* increase the disease resistance of plants by enhancing the activities of ROS scavenging enzymes and regulating ROS content. The results showed that when tobacco plants were infected with *Lo. piceae* the leaves of wild-type plants showed curls and yellowish disease spots, while the transgenic plants overexpressing *PaPR4-a* and *PaPR4-b* showed no abnormalities (Figure 9b). Additionally, the ROS scavenging enzymes activities of transgenic plants was 1.5–3.0 times higher than that of wild-type plants (Figure 9c); higher ROS scavenging enzymes activities could inhibit ROS accumulation and reduce oxidative damage. In addition, we detected the relative expression levels of *PaPR4-a* and *PaPR4-b* in transgenic plants after infection and found that their expression levels were both up-regulated by 6–7 times compared with those before infection (Figure 9d), which may be the reason for the stronger disease resistance of transgenic plants. However, the molecular mechanism of activating *PaPR4-a* and *PaPR4-b* transcription needs to be further studied. Previous studies have demonstrated that the existence of an oxidative burst-independent mechanism for the transcriptional activation of *PR* genes [70]. In subsequent studies, it was demonstrated that transgenic plants overexpressing transcription factor *SpWRKY* from tomato or peroxidase gene *CaPO2* from pepper had significantly higher expression levels of *PR* genes and lower content of ROS than wild-type plants after infection [71,72]. Therefore, the upstream genes that activate *PR4* gene transcription may be some transcription factors or peroxidases.

In conclusion, we cloned and identified two genes encoding Class II PR4 proteins, namely, *PaPR4-a* and *PaPR4-b*, in *Pi. asperata*. The recombinant proteins were expressed and purified in *E. coli*, and their antifungal activity was identified. Transgenic tobacco overexpressing *PaPR4-a* and *PaPR4-b* increased the activities of ROS scavenging enzymes and showed stronger disease resistance in the process of resistance to pathogenic fungi infection. These results indicated that *PaPR4-a* and *PaPR4-b* played an active role in the resistance of plants to pathogenic fungal infection and participated in the regulation of ROS network pathways. The findings highlight the significance of studying the functions of *PaPR4-a* and *PaPR4-b* genes and provides novel insights into the PR4 protein family of *Pi. asperata*. The study also serves as a reference for future studies of the PR4 family of proteins.

## 4. Materials and Methods

### 4.1. Plant Materials and Plasmids

The annual, *Pi. asperata*, used in this study was obtained from the Forest Pathology Laboratory of Sichuan Agricultural University.

The pMD19-T vector used for cloning and the pET-32a vector used for analyzing prokaryotic gene expression were purchased from TaKaRa Bio Inc. DH5α-competent and BL21 competent *E. coli* cells were used for cloning and protein expression, respectively, and were purchased from TransGen Biotech Co., Ltd. (Beijing, China). The pCAMBIA1300-EGFP-MCS vector and tobacco plants used for the subcellular localization studies were donated by the Forest Pathology Laboratory of Sichuan Agricultural University.

### 4.2. Total RNA Extraction and Reverse Transcription

Needles of *Pi. asperata* (100 mg) were ground with liquid nitrogen, and the total RNA was extracted using a FastPure^®^ Universal Plant Total RNA Isolation Kit (Nanjing Vazyme Biotech Co., Ltd., Nanjing, China). The concentration of RNA was determined using an ultramicro spectrophotometer (N60 Touch, Implen, Westlake Village, CA, USA). The synthesis of cDNA was performed according to the instructions provided in the PrimeScriptTM RT Reagent Kit (TaKaRa, Dalian, China). The cDNA was used for PCR amplification studies.

### 4.3. Cloning of Full-Length PaPR4-a and PaPR4-b Genes

Two specific primer pairs (PaPR4-a-F/PaPR4-b-R and PaPR4-b-F/PaPR4-b-R; Appendix A) were designed using the Premier software, version 5.0 (PREMIER Biosoft International, Palo Alto, CA, USA) based on the nucleotide sequences of the *PR4* gene of *Ps. menziesii* (GenBank IDs: JQ064526.1 and JQ064525.1). PCR amplification was performed using 2× Trans Taq^®^ High Fidelity (HiFi) PCR SuperMix (TransGen Biotech Co., Ltd., Beijing, China).

The PCR product was assessed by gel electrophoresis using 1% (M/V) agarose gels, and the target fragment was purified with a Universal DNA Purification Kit (Tiangen Biotech Co., Ltd., Beijing, China). The purified products were cloned into the pMD19-T vector (TaKaRa Bio Inc., Dalian, China) and then transformed into DH5α-competent *E. coli*. The transformed bacteria were screened by the blue/white screening technique and identified by PCR. A HighPure Plasmid Mini kit (Aidlab Biotechnologies Co., Ltd., Beijing, China) was used for plasmid extraction. The positive colonies were sequenced by Hangzhou Healthy Creatures Biotechnology Co., Ltd. (Hangzhou, China). The recombinant pMD-19T plasmids were stored at −20 °C in a refrigerator.

### 4.4. Bioinformatics Analyses of PaPR4-a and PaPR4-b

The open reading frames (ORFs) of PaPR4-a and PaPR-b were predicted using the ORF finder tool (http://www.bioinformatics.org/sms2/orf_find.html, accessed on 12 November 2021). The amino acid sequences of PaPR4-a and PaPR4-b were deduced using the DNAMAN software, version 6.0 (Lynnon, Pointe-Claire, QC, Canada). The sequence homologs of the identified PaPR4-a and PaPR4-b proteins were determined by BLAST search (http://blast.ncbi.nlm.nih.gov/Blast.cgi, accessed on 14 November 2021) against the National Center for Biotechnology Information (NCBI) database. The functional sites and domains were predicted using NCBI tools (https://www.ncbi.nlm.nih.gov/Structure/cdd/wrpsb.cgi, accessed on 14 November 2021). The physicochemical properties of the proteins were predicted using ExPASY (https://web.expasy.org/protparam/, accessed on 16 November 2021). The signal peptides were predicted using signalp-5.0 (https://services.healthtech.dtu.dk/service.php?SignalP-5.0, accessed on 17 November 2021), and the transmembrane regions were predicted using TMHMM-2.0 (https://services.healthtech.dtu.dk/service.php?TMHMM-2.0, accessed on 14 November 2021). The ProtScale webserver was used for predicting the hydrophilicity of the proteins (https://web.expasy.org/protscale/, accessed on 14 November 2021). The functional sites and domains were analyzed using NCBI tools (https://www.ncbi.nlm.nih.gov/Structure/cdd/wrpsb.cgi, accessed on 20 November 2021). The SOMPA tool was used for predicting the secondary structure of PaPR4-a and PaPR-b (https://npsa-prabi.ibcp.fr/cgi-bin/npsa_automat.pl?page=npsa_sopma.html, accessed on 2 December 2021). The tertiary structures of the proteins were predicted with SWISS-MODEL (https://swissmodel.expasy.org/interactive, accessed on 2 December 2021). Multiple sequence alignment was performed with ESpriPT3.0 (https://espript.ibcp.fr/ESPript/cgi-bin/ESPript.cgi, accessed on 2 December 2021). The phylogenetic tree was finally constructed with MEGA 6 [73], using the neighbor-joining method.

### 4.5. Expression of PaPR4-a and PaPR4-b in E. coli

Two primers pairs, EcoR I-PaPR4-a-F/EcoR I-PaPR4-a-R and EcoR I-PaPR4-b-F/EcoR I-PaPR4-b-R (Appendix A), devoid of signal peptides, were designed based on the sequences of the *PaPR4-a* and *PaPR4-b* genes. The ORFs of *PaPR4-a* and *PaPR4-b* were amplified from the cloned pMD-19T vector. A pair of validation primers, T7-F and T7-R (Appendix A), were designed at the two ends of the polyclonal site of the pET-32a vector for identifying the positive recombinant vector. The PCR products were detected by gel electrophoresis using 1% agarose gels, and the target fragment was purified. The pET-32a vector was digested using the *Eco*RI restriction endonuclease (New England Biolabs Ltd., Beijing, China) at 37 °C for 3 h, following which the linearized vector was purified and recovered. A Trelief^®^ Seamless Cloning Kit (Tsingke Biotechnology Co., Ltd., Beijing, China) was used for linking the target fragment to the linearized vector at 50 °C for 15 min. The pET-32a-PaPR4-a and pET-32a-PaPR4-b recombinant plasmids were transformed into *E. coli* BL21 (DE3) competent cells. The cells were then evenly coated on LB media containing 100 μg/mL ampicillin and cultured overnight at 37 °C. A single colony was selected the following day for PCR amplification. The positive colonies were isolated into liquid LB media containing 100 μg/mL ampicillin and cultured overnight at 180 rpm, at 37 °C. The plasmids were subsequently extracted and sequenced. The bacterial solution was mixed with 1:1 glycerol and stored in a −80 °C refrigerator.

The solution of positive bacteria was mixed with liquid LB medium containing 100 μg/mL ampicillin at a ratio of 1:100 and cultured at 180 rpm at 37 °C until the OD_600_ of the bacteria solution reached 2.5. In order to determine the optimal concentration of IPTG for induction, 1M IPTG was added to the bacterial solution to achieve final IPTG concentrations of 0, 0.2, 0.4, 0.6, 0.8 and 1.0 mM, and the bacteria were cultured at 180 rpm at 37 °C for 3 h. Then 1 mL of the bacterial solution was centrifuged at 13,000× *g* for 1 min and the supernatant was discarded. The protein expression levels were subsequently determined by SDS-PAGE using 12% gels for determining the optimal concentration of IPTG for induction. Bacterial solutions induced by the optimal concentration of IPTG were cultured at 37 °C at 180 rpm for 0, 1, 2, 3, 4 and 5 h, following which 1 mL of bacterial solution was collected from each experimental setup. The protein expression level was determined with SDS-PAGE using 12% gels and the optimal duration of induction was determined. Bacterial solutions containing the optimal IPTG concentration were cultured at 180 rpm at 20 °C, 180 rpm at 25 °C, 180 rpm at 30 °C and at 180 rpm at 37 °C, and 1 mL of bacterial solution was collected from each experimental setup. The bacteria were resuspended in 1× PBS buffer (Solarbio Biotechnology Co., Ltd., Beijing, China). The resuspended bacteria were sonicated on ice with an ultrasonic cell crusher (power 40 W, 4 s working, 6 s interval, for 15 min). The optimal induction temperature and solubility were determined by measuring the protein expression levels with SDS-PAGE using 12% gels.

#### 4.5.1. Washing of Inclusion Bodies

The method described in Section 4.5 was used for ultrasonic fragmentation of IPTG-induced bacteria and collecting the precipitate. The precipitate was added to 10 mL of pre-cooled washing solution (20 mmol/L Tris-HCl, 0.5 mol/L NaCl, 2 mol/L urea and 2% Triton X-100), stirred for 20 min and centrifuged at 13,000× *g* for 25 min at 4 °C, following which the supernatant was discarded and the process was repeated. The precipitate was washed with 50 mmol/L Tris-HCl once more, following which the centrifugation was repeated for collecting the precipitate.

#### 4.5.2. Dissolution of Inclusion Bodies

The precipitate was suspended with 10 mL solution of 20 mmol/L Tris-HCl, 0.5 mol/L NaCl, 6 mol/L urea, 1 mmol/L β-mercaptoethanol and 2% TritonX-100, after which the mixture was stirred at 25 °C for 1 h, centrifuged at 13,000× *g* for 30 min at 4 °C, and the precipitate was discarded. The supernatant was collected and filtered using a 0.22 μm bacterial filter.

#### 4.5.3. Renaturation and Purification of PaPR4-a and PaPR4-b

HisSep Ni-NTA Agarose Resin (Yeasen Biotechnology Co., Ltd., Shanghai, China) was activated using 5–10 mL of buffer solution (20 mmol/L Tris-HCl, 500 mmol/L NaCl, 6 mol/L urea, 5 mmol/L imidazole, and 1 mmol/L β-mercaptoethanol). The previously obtained supernatant solution was added to the column. The column was cleaned with 10 mL of washing buffer (20 mmol/L Tris-HCl, 500 mmol/L NaCl, 6 mol/L urea, 5 mmol/L imidazole, and 1 mmol/L mercaptoethanol). The column was then cleaned with an eluent (20 mmol/L Tris-HCl, 500 mmol/L NaCl, 0.1 mmol/L oxidized glutathione and 1 mmol/L reduced glutathione) containing 8, 6, 4, 2 and 0 mol/L urea. The column was subsequently cleaned with an eluent containing 400 mmol/L imidazole. The eluent obtained from each stage was collected and analyzed by sodium dodecyl sulphate polyacrylamide gel electrophoresis (SDS-PAGE) using the method described in the Section 3.

### 4.6. Antifungal Activity of Recombinant PaPR4-a and PaPR4-b Proteins

The antifungal activity of the recombinant PaPR4-a and PapPR4-b proteins was determined by the alterations in the mycelial morphology of *Lo. piceae*. To this end, 200 μL of purified recombinant protein and a fungus cake with a diameter of 0.8 cm were placed in a clean 1.5 mL centrifuge tube. The tube was then placed in an incubator at 25 °C. Mycelial morphology was observed under a microscope after 24 h. The experiments were performed in triplicate, and the eluent was used as the control.

### 4.7. Subcellular Localization of PaPR4-a and PaPR4-b

The EGFP-PaPR4-a-F/EGFP-PaPR4-a-R and EGFP-PaPR4-b-F/EGFP-PaPr4-b-R primer pairs (Appendix A) were designed based on the gene sequences of *PaPR4-a* and *PaPR4-b* for constructing the subcellular localization vectors. The ORFs of PaPR4-a and PaPR4-b were amplified from the cloned pMD-19T vector. A pair of validation primers, BamHI-yz-F and BamHI-yz-R (Appendix A), were designed based on the sequences of the ends of the polyclonal site of the pCAMBIA1300-EGFP-MCS vector for identifying the positive recombinant vector. The PCR products were detected by gel electrophoresis using 1% agarose gels, and the target fragment was purified. The pCAMBIA1300-EGFP-MCS vector was digested for 3 h at 37 °C with the *Bam*HI restriction endonuclease (New England Biolabs Ltd., Beijing, China), following which the linearized vector was purified and recycled and subsequently linked, transformed and preserved according to the method described previously. The EGFP-PaPR4-a and EGFP-PaPR4-b recombinant plasmids had kanamycin resistance genes. The bacteria were therefore cultured at 28 °C for 48 h in media containing 100 μg/mL kanamycin.

The EGFP-PaPR4-a and EGFP-PaPR4-b recombinant plasmids were transferred into *Ag*.-competent GV3101 cells according to the method described by Sparkes et al. [74] and used for infecting tobacco plants grown for 4 weeks (tobacco growing conditions: 25 °C, 14 h/10 h light/dark period and relative humidity of 70%). The infected tobacco plants were incubated in an incubator at 25 °C for 2 days and observed under a confocal laser microscope (Olympus FV3000).

### 4.8. Pathogens and Inoculation Procedures

*Lo. piceae* was maintained in the Sichuan Agricultural University Culture Collection (SICAUCC) and was isolated from Sichuan Province, China. *Lo*. *piceae* strains were cultured at 25 °C in PDA medium (200 g/L potato 20 g/L agar and 20 g/L glucose). Plants were infected with 0.6 cm diameter fungus cake and moisturized, and the inoculation procedure was completed.

Needles of *Pi. asperata* were infected for 0, 24, 48, 72, 96, 120, 144 and 168 h, and the total RNA was extracted using the method described in the Section 3. Reverse transcription was performed with Hifair^®^V one-step RT-GDNA Digestion SuperMix for qPCR (Yeasen Bio Co., Ltd., Shanghai, China). Two primer pairs, PaPR4-a-Fq/PaPR4-a-Rq and PaPR4-b-Fq/PaPR4-b-Rq (Appendix A), were designed for RT-qPCR based on the sequences of *PaPR4-a* and *PaPR4-b*. *EF* (GenBank ID: AJ132534.1) and *TIF* (GenBank ID: DR448953.1) were used as the internal reference genes (Appendix A) [75,76,77]. Hieff UNICON^®^ Universal Blue qPCR SYBR Green Master Mix (Yeasen Biotechnology Co., Ltd., Shanghai, China) was used for RT-qPCR, and the experiments were performed in triplicate. The relative gene expression levels were calculated using the 2^−∆∆Ct^ method [78], and significant differences in gene expression were analyzed using IBM SPSS Statistics 20 (SPSS Inc., Chicago, IL, USA), while GraphPad Prism 7 (GraphPad Software, San Diego, CA, USA) was used for mapping the data.

### 4.9. Functional Verification of PaPR4-a and PaPR4-b

The OE-PaPR4-a-F/OE-PaPR4-a-R and OE-PaPR4-b-F/OE-PaPr4-b-R primer pairs (Appendix A) were designed based on the gene sequences of *PaPR4-a* and *PaPR4-b* for constructing the overexpression vectors. The ORFs of PaPR4-a and PaPR4-b were amplified from the cloned pMD-19T vector. The PCR products were detected by gel electrophoresis using 1% agarose gels, and the target fragment was purified. The BG Plant-Express MCS vector was digested for 3 h at 37 °C with the *Sma I* restriction endonuclease (New England Biolabs Ltd., Beijing, China), following which the linearized vector was purified and recycled and subsequently linked, transformed and preserved according to the method described previously. 35S-F and NOST-R primers (Appendix A) were used as validation primers. The OE-PaPR4-a and OE-PaPR4-b recombinant plasmids had spectinomycin resistance genes. The bacteria were therefore cultured at 28 °C for 48 h in media containing 100 μg/mL spectinomycin. The positive recombinant plasmid OE-PaPR4-a and OE-PaPR4-b were transformed into *Ag*.-competent GV3101 cells, which was then transformed into *N. benthamiana* using the leaf disk method described by Oh et al. [79] Transgenic plants were obtained for subsequent experiments (Appendix A). The OE-PaPR4-a-F/OE-PaPR4-a-R and OE-PaPR4-b-F/OE-PaPr4-b-R primer pairs were used to amplify the target bands from the genomes to detect transgenic plants.

Tobacco plants were infected using the methods described above. Leaves of wild-type and transgenic tobacco were infected for 0, 24, 48, 72 and 96 h, and their SOD, POD, and CAT activities were measured using the superoxide dismutase (SOD) kit (Grace Biotechnology Co., Ltd., Suzhou, China), the peroxidase (POD) kit (Grace Biotechnology (Suzhou, China) Co., Ltd.) and the catalase (CAT) Kit (Grace Biotechnology Co., Ltd., Suzhou, China). For tobacco plants, the *PP2A* and *F-BOX* genes from *N. benthamiana* was amplified and used to normalize the values as an internal control (Appendix A) [80] and the experiments were performed in triplicate.

## Figures and Tables

**Figure 1 ijms-23-14906-f001:**
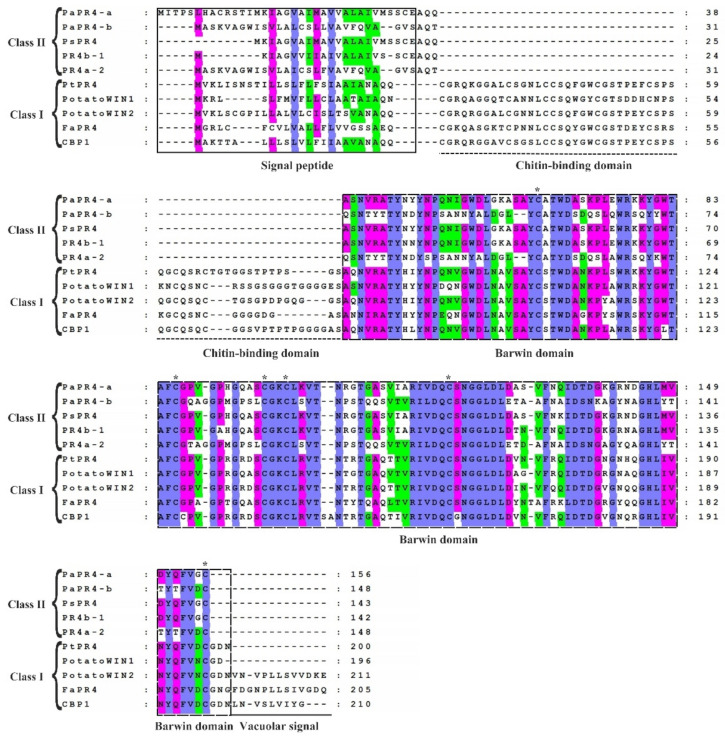
Comparison of the sequences of Class I and Class II PR4 proteins. The sequences of Class I proteins included PtPR4 (*Populus trichocarpa*; GenBank ID: XP_002319077.1), PotatoWIN1 (*Solanum tuberosum* L.; GenBank ID: P09761.1), PotatoWIN2 (*So. tuberosum* L.; GenBank ID: P09762.1), FaPR4 (*Ficus pumila* var. awkeotsang; GenBank ID: ADO24163.1), and CBP1 (*Capsicum annuum*; GenBank ID: AAF18934.1). The sequences in Class II proteins included PaPR4-a, PaPR4-b, PsPR4 (*Picea sitchensis*; GenBank ID: ABK23104.1), PR4b-1 (*Pseudotsuga menziesii*; GenBank ID: AFD50744.1) and PR4a-2 (*Ps*. *menziesii*; GenBank ID: AFD50743.1). The signal peptide, chitin-binding domain, Barwin domain, and vacuolar signal are indicated. The identical, highly similar and similar amino acids are indicated in blue, pink and green, respectively. Six cysteines which are highly conserved within the Barwin domain are indicated by asterisks (*).

**Figure 2 ijms-23-14906-f002:**
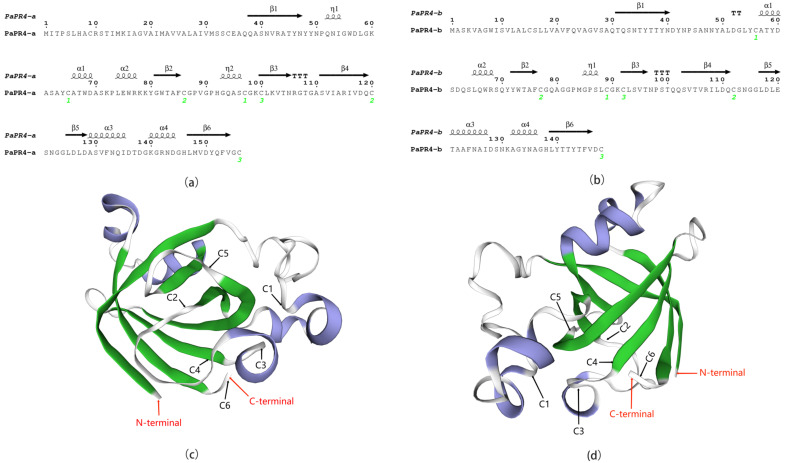
Structural analysis of PaPR4-a and PaPR4-b. (**a**,**b**) represent the secondary structures of PaPR4-a and PaPR4-b. The α-helices, η-helices, β-chains and TTT are indicated. The numbers in green represent the positions of the three disulfide bonds. (**c**,**d**) represent the tertiary structures of PaPR4-a and PaPR4-b constructed using the Barwin-like protein of papaya as template. The α -helices and β-chains are indicated in blue and green, respectively, while the red arrows indicate the N-termini and C-termini of the proteins. C1–C6 indicate the six cysteines that formed three disulfide bonds in the proteins.

**Figure 3 ijms-23-14906-f003:**
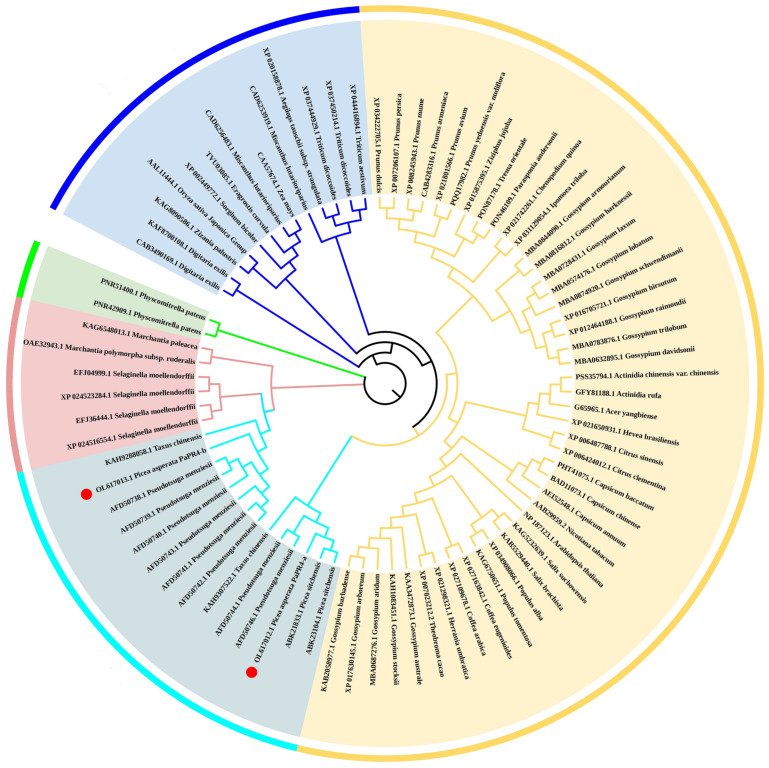
Phylogenetic tree of PR4 proteins constructed using the adjacency method. The monocotyledons, dicotyledons, gymnosperms, ferns and bryophytes are indicated in blue, yellow, cyan, pink and green, respectively. PaPR4-a and PaPR4-b are indicated as ●.

**Figure 4 ijms-23-14906-f004:**
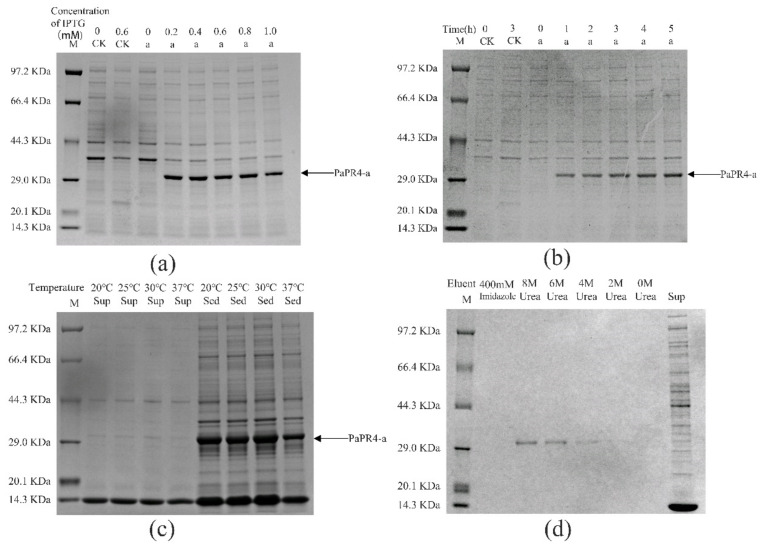
Analysis of the recombinant PaPR4-a protein with SDS-PAGE. M: Protein marker. CK: pET-32a vector expressed in *E. coli* BL21 (DE3) cells. Sup: supernatant. Sed: sediment. (**a**) The recombinant PaPR4-a protein was induced to express at 0, 0.2, 0.4, 0.6, 0.8 and 1.0 mM concentrations of IPTG. (**b**) Expression levels of the recombinant PaPR4-a protein at 0, 1, 2, 3, 4 and 5 h. (**c**) Induction of expression and detection of solubility of the recombinant PaPR4-a protein at 20 °C, 25 °C, 30 °C and 37 °C. (**d**) Renaturation and purification of the recombinant PaPR4-a protein.

**Figure 5 ijms-23-14906-f005:**
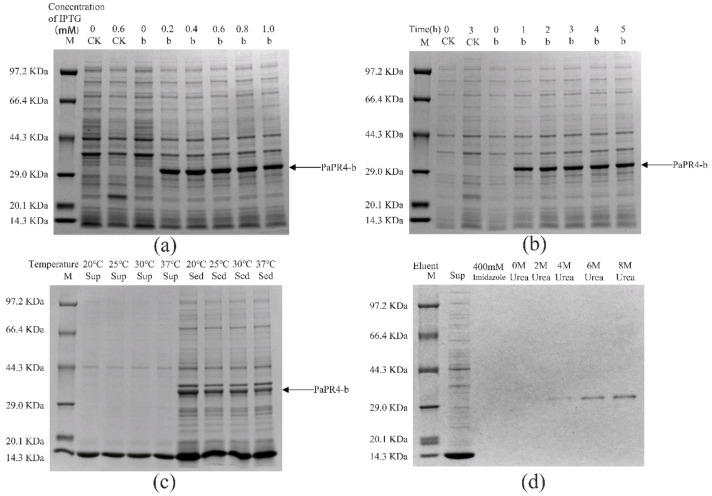
Analysis of the recombinant PaPR4-b protein with SDS-PAGE. M: Protein marker. CK: pET-32a vector expressed in *E. coli* BL21 (DE3) cells. Sup: supernatant. Sed: sediment. (**a**) The recombinant PaPR4-b protein was induced to express at 0, 0.2, 0.4, 0.6, 0.8 and 1.0 mM concentrations of IPTG. (**b**) Expression levels of the recombinant PaPR4-b protein at 0, 1, 2, 3, 4 and 5 h. (**c**) Induction of expression and detection of solubility of the recombinant PaPR4-b protein at 20 °C, 25 °C, 30 °C and 37 °C. (**d**) Renaturation and purification of the recombinant PaPR4-b protein.

**Figure 6 ijms-23-14906-f006:**
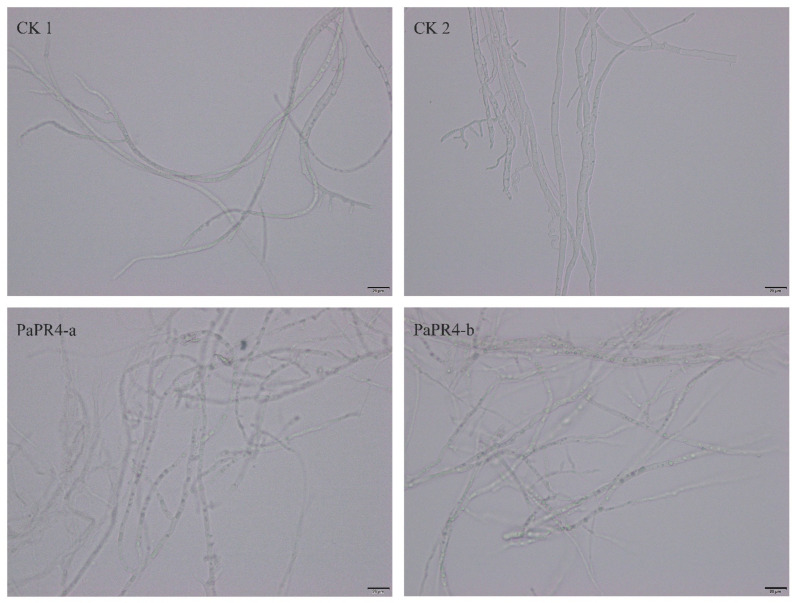
Antifungal activity of recombinant PaPR4-a and PaPR4-b proteins.CK1 and CK2 represent mycelia treated with sterile water and eluent, respectively, while PaPR4-a and PaPR4-b represent the experimental groups. Each experiment was performed in triplicate. The scale bar in the figure is 20 μm.

**Figure 7 ijms-23-14906-f007:**
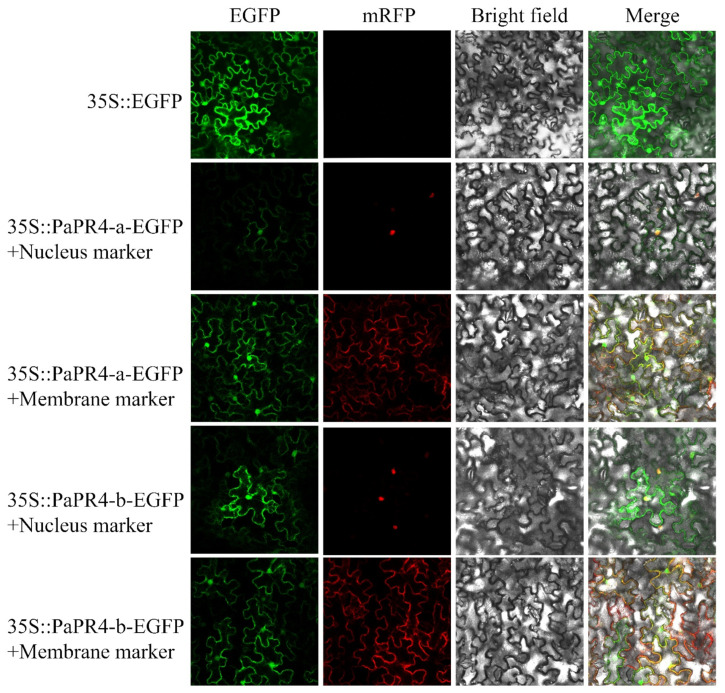
Subcellular localization of PaPR4-a and PaPR4-b in tobacco leaves. 35S:EGFP represents the control group.

**Figure 8 ijms-23-14906-f008:**
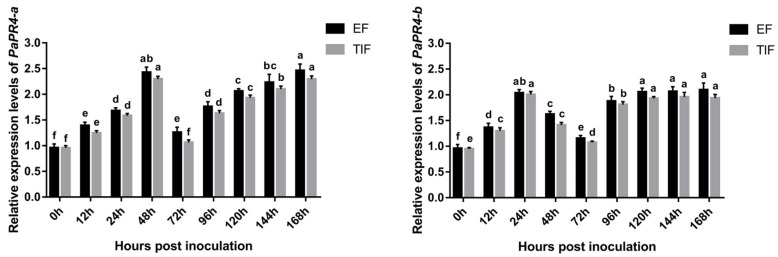
The expression levels of *PaPR4-a* and *PaPR4-b* at different durations after the infection of *Pi. asperata* with *Lo. piceae*. The relative expression levels represent the normalized mean ± standard deviation (SD) (*n* = 3 biological replicates). The different letters indicate significant differences between samples determined by the Waller-Duncan test (*p* < 0.05).

**Figure 9 ijms-23-14906-f009:**
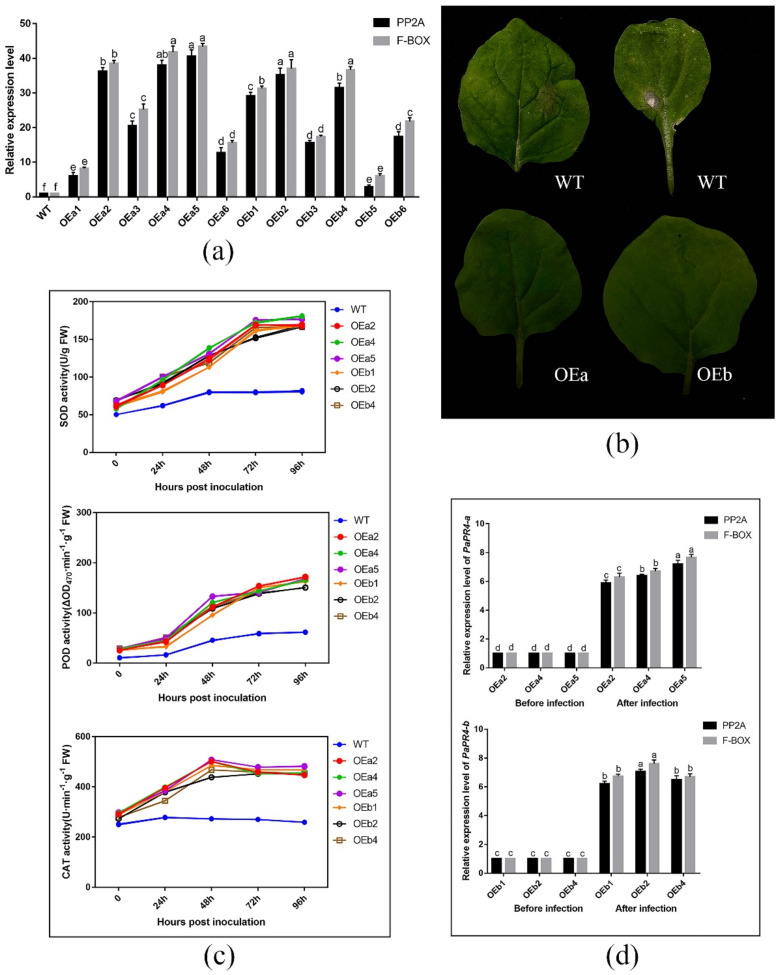
(**a**) RT-qPCR detection of *PaPR4-a* and *PaPR4-b* expression in twelve transgenic lines; (**b**) Phenotype of wild-type and transgenic tobacco plants inoculated with *Lo. piceae* at a week post-inoculation; (**c**) ROS scavenging enzymes activities after inoculation; (**d**) Relative expression levels of *PaPR4-a* and *PaPR4-b* after inoculation. WT: wild type; FW: fresh weight. The relative expression levels represent the normalized mean ± standard deviation (SD) (*n* = 3 biological replicates). The different letters indicate significant differences between samples determined by the Waller-Duncan test (*p* < 0.05).

## Data Availability

Not applicable.

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
