# Peer review of "Cloning and Characterization of Two Novel PR4 Genes from Picea asperata"

_ijms, 2022, doi:10.3390/ijms232314906_

Round 1

Reviewer 1 Report

In this study, two PR4 genes, PaPR4-a and PaPR4-b, from Picea asperata, were isolated, cloned, and expressed in different organisms. Also extensive in silico analyzes were performed. Although there are a few problems with the text, it can be accepted for publication as it is.

Suggestions

1. Revise the sentences located on line 101, line 77 and line 25

Author Response

Dear reviewer,

Thank you for your comments concerning our manuscript entitled “ijms-1918432”. Those comments are all valuable and very helpful for revising and improving our paper, as well as the important guiding significance to our researches. We have studied comments carefully and have made correction which we hope meet with approval. Revised portion are marked in red in the paper. The main corrections in the paper and the responds to the reviewer’s comments are as flowing:

Point 1: Revise the sentences located on line 101, line 77 and line 25.

Response 1: The modification has been completed (L24-25 and L101-102). Please see the attachment.

Reviewer 2 Report

In the current study, the authors isolated two PR4 genes, PaPR4-a and PaPR4-b, from Picea asperata, induced their expression in E. coli, and investigated the antifungal activities of the corresponding purified proteins. Authors also performed in silico studies of these two proteins, and their sub-cellular localization, and also analyzed the transcript patterns of two genes after inoculation of Pi. asperata with Lo. Piceae fungus.

The manuscript at this stage appears very primitive and substantial additions are required to be considered for the next level. Some of the major concerns are highlighted below:

1.       The PaPR4-a and PaPR4-b genes should have been overexpressed in the native Picea asperata system and their antifungal activities on high-expressing Pi. asperata plants would have provided insights into whether these candidates are indeed involved in inducing resistance. At least transient overexpression followed by fungal responsiveness should be carried out.

2.       Did the authors check the antifungal activity on plants? I recommend they check the antifungal activity on Pi. asperata.

3.       Why PaPR4-a and PaPR4-b existed as inclusion bodies in E. coli? Is it because of protein misfolding in a non-host system like bacteria? Or do they exist as inclusion bodies in native cells also?

4.       The upregulation of transcripts of both PR4a and PR4b post inoculation was around 2 to 2.5-fold which is not very high. Is the inoculation and transcript analysis properly performed?

5.    Materials & Methods are not described in detail, for instance, the methodology of inoculation is not mentioned properly.

6.       In the Discussion (Line 283-284), the authors stated that E. coli continues to be the preferred system for fungal interaction studies and provided old references from 1999 and 2003. With the advancements in plant genetic engineering, breakthroughs have been made in the genetic transformation of recalcitrant plant species and their molecular characterizations. This statement should be modified and Lo. Piceae inoculation studies on high expression lines of Pi. asperata must be conducted.

7.       English proficiency must be thoroughly improved.

Author Response

Dear reviewer,

Thank you for your comments concerning our manuscript entitled “ijms-1918432”. Those comments are all valuable and very helpful for revising and improving our paper, as well as the important guiding significance to our researches. We have studied comments carefully and have made correction which we hope meet with approval. Revised portion are marked in red in the paper. The main corrections in the paper and the responds to the reviewer’s comments are as flowing:

Point 1: The PaPR4-a and PaPR4-b genes should have been overexpressed in the native Picea asperata system and their antifungal activities on high-expressing Pi. asperata plants would have provided insights into whether these candidates are indeed involved in inducing resistance. At least transient overexpression followed by fungal responsiveness should be carried out.

Response 1: The function of the PaPR4-a and PaPR4-b was further investigated by overexpression experiment. However, regeneration system of tissue culture through callus has not been established due to the growth characteristics of the Picea asperat itself, the PaPR4-a and PaPR4-b were overexpressed in Nicotiana benthamiana respectively and their biological function in defending against Lophodermium piceae were analyzed. The experimental content has been added to the manuscript. Please see the attachment.

Point 2: Did the authors check the antifungal activity on plants? I recommend they check the antifungal activity on Pi. asperata.

Response 2: Because of the growth characteristics of Pi. asperata, The antifungal activity was verified in transgenic tobacco, which showed that the ROS scavenging enzymes activities of transgenic tobacco were increased after Lo. Piceae infection, and the transgenic tobacco had stronger disease resistance. The experimental content has been added to the manuscript. Please see the attachment.

Point 3: Why PaPR4-a and PaPR4-b existed as inclusion bodies in E. coli? Is it because of protein misfolding in a non-host system like bacteria? Or do they exist as inclusion bodies in native cells also?

Response 3: The existence of recombinant proteins in inclusion bodies may be caused by various reasons such as culture conditions, expression vectors and characteristics of Escherichia coli. There are many recombinant PR proteins present in the form of inclusion bodies. Some studies have shown that N-terminal amino acids sequences of soluble proteins shared high sequence similarities with tobacco thaumatin-like protein(doi:10.1093/abbs/36.11.773). After comparison, we guessed that PaPR4-a and PaPR4-b may belong to insoluble proteins. This result has little impact on the rest of our study.

Point 4: The upregulation of transcripts of both PR4a and PR4b post inoculation was around 2 to 2.5-fold which is not very high. Is the inoculation and transcript analysis properly performed?

Response 4: It was confirmed that the inoculation and transcript analysis were correct. The reason may be the low vitality of the pathogen used at that time or the ambient temperature. We reisolated new pathogens (Lophodermium piceae) for subsequent experiments, and the relative expression of PaPR4-a and PaPR4-b increased 6-7 times in the infected transgenic tobacco. The experimental content has been added to the manuscript. Please see the attachment.

Point 5: Materials & Methods are not described in detail, for instance, the methodology of inoculation is not mentioned properly.

Response 5: Inoculation methods have been added to the materials and methods (4.8. Pathogens and Inoculation Procedures). The experimental content has been added to the manuscript. Please see the attachment.

Point 6: In the Discussion (Line 283-284), the authors stated that E. coli continues to be the preferred system for fungal interaction studies and provided old references from 1999 and 2003. With the advancements in plant genetic engineering, breakthroughs have been made in the genetic transformation of recalcitrant plant species and their molecular characterizations. This statement should be modified and Lo. Piceae inoculation studies on high expression lines of Pi. asperata must be conducted.

Response 6: The sentence has been revised (Line 331-332). The gene function verification experiments of overexpressing PaPR4-a and PaPR4-b have been completed and the experimental content has been added to the manuscript. Please see the attachment.

Point 7: English proficiency must be thoroughly improved.

Response 7: The manuscript has been edited by a native English editor. The editorial certificate is in the attachment.炜栋 赵is the Chinese writing of Wei-Dong Zhao.

Reviewer 3 Report

The MS entitled “Cloning and characterization of two novel PR4 genes from Picea asperata” has been reviewed. Authors conducted the homolog clone of two pathogenesis-related proteins, then performed a set of bioinformatic analyses and basic validation. The results could provide the basis for further research on PR4-induced defense responses e in Pi. asperata. Overall, the MS need major revision, and to clarify some concerns as addressed, especially re-write the results part.

1.      There should more studies in model plant on different type CBD and PR protein, should list more information and discuss.

2.      Why did you choose the PaPR4-a and PaPR4-b instead of others? should give some demonstration.

3.      The author may delete some of bioinformatics analyses, and adding more on functional analysis.

4.      L162-170, L180-186, L188-192, L194-203, L233-245, and L252-258, these parts were quite similar with the methods, or summary of the methods. Here is the result part, should list the results what your found, analysis the results, and describe the innovation.

5.      More validation work should confirm in vivo to setup the conculsion, although may not add in this research.

6.      Should discuss more about the innovation with other research.

7.      English should carefully check through the MS.

Author Response

Dear reviewer,

Thank you for your comments concerning our manuscript entitled “ijms-1918432”. Those comments are all valuable and very helpful for revising and improving our paper, as well as the important guiding significance to our researches. We have studied comments carefully and have made correction which we hope meet with approval. Revised portion are marked in red in the paper. The main corrections in the paper and the responds to the reviewer’s comments are as flowing:

Point 1: There should more studies in model plant on different type CBD and PR protein, should list more information and discuss.

Response 1: Many PR genes have been reported in model plants, but more model plants were used to verify the function of the genes. Therefore, we supplemented the gene function verification experiment The functions of PaPR4-a and PaPR4-b were further investigated by overexpression in Nicotiana benthamiana. The experimental content has been added to the manuscript. Please see the attachment.

Point 2: Why did you choose the PaPR4-a and PaPR4-b instead of others? should give some demonstration.

Response 2: Because studies on gymnosperms have reported genes of the PR4 family, namely, PmPR4a1, PmPR4a2, and PmPR4b1, in Douglas fir, and their expression levels have been shown to be upregulated following infection with Phellinus sulphurascens. Based on the known sequences, we successfully cloned PaPR4-a and PaPR4-b and other genes. The size of genes PaPR4-a and PaPR4-b were suitable for subsequent experiments, and their relative expression levels were upregulated after infection.

Point 3: The author may delete some of bioinformatics analyses, and adding more on functional analysis.

Response 3: The gene function verification experiments of overexpressing PaPR4-a and PaPR4-b have been completed and the experimental content has been added to the manuscript. Please see the attachment.

Point 4: L162-170, L180-186, L188-192, L194-203, L233-245, and L252-258, these parts were quite similar with the methods, or summary of the methods. Here is the result part, should list the results what your found, analysis the results, and describe the innovation.

Response 4: We have modified the Materials Methods and Results sections and the modified results has been added to the manuscript. Please see the attachment.

Point 5: More validation work should confirm in vivo to setup the conculsion, although may not add in this research.

Response 5: PaPR4-a and PaPR4-b functional validations have been added to the manuscript. The experimental content has been added to the manuscript. Please see the attachment.

Point 6: Should discuss more about the innovation with other research.

Response 6: We cloned and identified PaPR4-a and PaPR4-b from Picea asperat, explored their antifungal activity, and further verified the function of the gene through overexpression experiment. To date, this study is the first to report the identification of PR4 proteins of Pi. asperata and provides novel insights into the PR4 protein family of gymnosperms

Point 7: English should carefully check through the MS.

Response 7: The manuscript has been edited by a native English editor. The editorial certificate is in the attachment.炜栋 赵is the Chinese writing of Wei-Dong Zhao.

Round 2

Reviewer 3 Report

Overall, the MS improved a lot than the 1st edition.

Should add more description on new figure9.

L306, some puntuation mark were in Chinese style.

Author Response

Dear reviewer,

Thank you for your comments concerning our manuscript entitled “Cloning and characterization of two novel PR4 genes from Picea asperata”. The manuscript ID is “ijms-1918432”. Those comments are all valuable and very helpful for revising and improving our paper, as well as the important guiding significance to our researches. We have studied comments carefully and have made correction which we hope meet with approval. Revised portion are marked in red in the paper. The main corrections in the paper and the responds to the reviewer’s comments are as flowing:

Point 1: English language and style are spell check required

Response 1:We have checked the English language and style

Point 2: Should add more description on new figure9.

Response 2: We have described more about the gene function verification section and Figure 9(L320-332).

Point 3: L306, some puntuation mark were in Chinese style.

Response 3: The modification has been completed (L306). Please see the attachment.
